# Learning-based Synthesis of Social Laws in STRIPS

## Ronen Nir, Alexander Shleyfman, Erez Karpas

Technion — Israel Institute of Technology

ronenn@campus.technion.ac.il, shleyfman.alexander@gmail.com, karpase@technion.ac.il

## Abstract

In a multi-agent environment, each agent must take into account not only the actions it must perform to achieve its goals, but also the behavior of other agents in the system, which usually requires some sort of coordination between the agents. One way to avoid the complexity of centralized planning and online negotiation between agents is to design an artificial social system. This system enacts a social law that restricts the behavior of the agents. A robust social law enables the agents to reach their goals while keeping them from interfering with each other. However, the problem of efficient synthesis of such laws is computationally hard, and previously proposed search techniques do not scale well. In this paper, we propose the use of graph neural networks to predict social laws from a graph-based representation of multi-agent systems. However, as this prediction can be wrong, we use heuristic search to correct possible mistakes in the network's prediction ensuring that the produced social law is indeed robust. Our empirical evaluation shows that this approach beat the previous state-of-the-art in social law synthesis and that it can learn from an imperfect expert, even in the presence of noise.

## Introduction

Multi-agent systems are becoming increasingly more common, with applications such as warehouse order fulfillment, autonomous cars, drone delivery, and more. In this paper, we focus on multi-agent systems with classical planning capabilities, known as multi-agent planning (MAP) (Borrajo and Fernández 2019). Automated MAP solves the problem of generating sequences of actions that achieve the individual goals of the agents starting from a given initial state. The design of such agents and environments is quite different from the design of an isolated agent and should take into account conflicts between agents, communication, privacy, *etc.* In a multi-agent setting, a plan that would have allowed an agent to obtain its goals in solitude may yield unexpected consequences. As Helmut von Moltke put it "no plan survives contact with the enemy".

Broadly speaking, the approaches to solve multi-agent planning can be divided into two categories: centralized and distributed, where each of the approaches has its drawbacks (Wooldridge 2009; Shoham and Leyton-Brown 2009). Centralized planning methods, usually, refer to the whole system as one planning problem, which neglects the privacy issues and usually do not scale well due to exponential growth in the number of agents. While being useful in numerous domains, this approach might suffer from limitations, such as the mentioned bottlenecks, or system-wide vulnerability to failure. On the other hand, the distributed approach requires some sort of coordination between the agents, where the coordination can be either implicit or explicit. Explicit coordination requires agents to negotiate (usually repeatedly) for conflict resolution, which requires time and physical embodiment of the communication devices. Implicit coordination, in turn, involves an element of system design that either imposes some rules of encounter on the agents or enacts a social law restricting the agents from performing certain actions under some conditions.

Our focus here is on the social law approach (Tennenholtz and Moses 1989), which is a form of implicit coordination. A social law restricts the allowed behaviors of the agents. A social law is called *robust* if, assuming all agents follow the law, they are all guaranteed to achieve their goals. Recent work (Nir, Shleyfman, and Karpas 2020) proposed an automatic method for synthesizing robust social laws for a given multi-agent environment. This is done by treating this problem as a heuristic search problem through the space of possible social laws. However, as this search space is huge (exponential in the number of actions), this technique does not scale well to large problems. Morales *et al.* (2018) proposed a technique of social law synthesis in the context of normative systems. Their approach is based on an evolutionary algorithm and does not guarantee completeness.

Theoretically, if we had a complete specification of the possible instances in a given MAP domain, we could use formal methods to synthesize a generic social law for the domain. However, MAP domains typically do not have such a specification, rather only a description of the schematic predicates and operators, without any specification of the possible combinations of initial states and goals (Shleyfman and Karpas 2018). Thus, the generalization of social laws, and the study of domain structures, though an interesting topic, can not provide guarantees about unseen problem instances.

Instead, we focus on a statistical machine learning approach, in which we generalize from labeled examples – social laws for MAP instances (MAPs), to social laws in

unseen MAPs of the same domain. Specifically, we assume access to a domain-expert, who provides us with a set of examples – small instances of the same domain, with a robust social law for each instance. We introduce a technique for learning from such examples and show that it is possible to generalize from such small examples and use the learned knowledge to synthesize social laws in large problem instances. Our aim here is to produce a robust social law for a given MAP based on the knowledge provided by the domain-expert. To do so, we use graph neural networks, combined with a search procedure to correct possible mistakes in the network's prediction, as a fast, domain-independent solution for the problem of social law generation.

Our technique works by representing multi-agent planning problems as graphs (extending the notion of Problem Description Graph (PDG) (Pochter, Zohar, and Rosenschein 2011) to the multi-agent setting). We train a Graph Convolutional Neural network (GCN) (Kipf and Welling 2017) on our labeled examples and use it to predict social laws on larger problems. However, as there are no guarantees that the predicted social law will be robust, we use the network output to speed up the search for social laws (Nir, Shleyfman, and Karpas 2020) by (a) introducing new heuristics based on the prediction, and (b) seeding the open list with potential social law candidates.

We empirically show that our technique can exploit a few small labeled examples to synthesize social laws for larger MAPs in the same domain, even when the domain-expert is imperfect. We also address the drawbacks of this approach, i.e., the domain-expert may not be able to locate all of the possible conflicts or produce a fully robust social law for each of the proposed examples. This is simulated by introducing noise to the social laws produced by the expert. We show that our approach is noise-tolerant, and can sometimes even benefit from such noise.

## Preliminaries

In this work we consider a multi-agent planning (MAP) setting that reasons about multiple non-collaborative, non-communicating agents. We start with the definition of the modified version of MA-STRIPS (Brafman and Domshlak 2008) introduced by Karpas *et al.* (2017). This modification employs individual goals for agents rather than a single goal for the entire problem.

Formally, a MAP problem in this setting is defined as a tuple $\Pi = \langle F, \{A_i\}_{i=1}^n, I, \{G_i\}_{i=1}^n \rangle$, where: $n \in \mathbb{N}$ represents the number of agents, $F$ is a set of facts, $I \subseteq F$ is the initial state, and $G_i \subseteq F$ and $A_i$ define the goal and the actions of agent $i$, correspondingly. The set of all actions of the problem and the overall goal of the problem are written as $A = \bigcup_{i=1}^n A_i$ and $G = \bigcup_{i=1}^n G_i$. We say that $s$ is a state of the system $\Pi$ if $s \subseteq F$, and $S = 2^F$ is the set of all states of $\Pi$. Each action $a \in A_i$ is described by preconditions $\mathsf{pre}(a) \subseteq F$, negative preconditions $\mathsf{npre}(a) \subseteq F^1$, add effects $\mathsf{add}(a) \subseteq F$, and delete effects $\mathsf{del}(a) \subseteq F$. We

---

[1]For domains with no negative preconditions we assume $\mathsf{npre}(a) = \emptyset$.

say that the action $a$ is well-defined if $\mathsf{pre}(a) \cap \mathsf{npre}(a) = \mathsf{add}(a) \cap \mathsf{del}(a) = \emptyset$. The action $a$ is applicable in the state $s$ if $\mathsf{pre}(a) \subseteq s$ and $s \cap \mathsf{npre}(a) = \emptyset$. The result of such application is be denoted by $s[\![a]\!] := (s \setminus \mathsf{del}(a)) \cup \mathsf{add}(a)$. In what follows we assume well-defined, unit-cost actions.

We define a projection of $\Pi$ for an agent $i$ to be $\Pi_i = \langle F, A_i, I, G_i \rangle$. Note that $\Pi$ is a regular (single-agent) STRIPS task (Fikes and Nilsson 1971). A sequence of actions $\pi_i$ is called a plan for the task $\Pi_i$ if the actions are consequently applicable starting from the state $I$ and resulting in a state $s$, s.t. $G_i \subseteq s$. The single-agent plans in the MAP context are referred to as *individual plans*. Following, Fišer et al. (2019), for an action set $L \subseteq A$, we denote $\Pi_i \setminus L := \langle F, A_i \setminus L, I, G_i \rangle$, generalizing it to the MAP setting as $\Pi \setminus L = \langle F, \{A_i \setminus L\}_{i=1}^n, I, \{G_i\}_{i=1}^n \rangle$.

We consider an execution model where each agent plans its actions in advance, and a given arbitrary scheduler executes the actions one by one with no given priorities, i.e., the scheduler can execute the plans provided by the agents in *any* possible order. At each step of the joint plan execution, the agent chosen by the scheduler acts according to its pending action, changing the state of the world accordingly. The scheduler repeats this procedure until one of the following happens: either all individual plans are executed, or some agent fails to execute its pending action. This failure to execute an action is denoted as a *conflict*. In what follows, we consider the scheduler to be adversarial, i.e., we would like to avoid any possible joint execution that results in either a conflict or in reaching a state $s$ where one of the agents has not achieved its goal (for some $i$ it holds that $G_i \not\subseteq s$).

To ensure that no execution results in such a failure, Karpas *et al.* introduced the notion of *social law* for MAP[2] – a set restrictions on agents' behavior that are meant to prevent conflicts. They described a social law as a modification to a MAP task $\Pi$ that can add, delete or modify facts, actions, the initial state, or the goal, and showed that some restrictions require complex conditions, for which the original set of facts is insufficient. In this work, we follow in the footsteps of Nir *et al.* (2020), which concentrated only on social laws that restrict conflict-inducing actions, while preserving the reachability of all agent goals, without altering the initial set of facts.

### Robust Social Laws

In this work we follow the definition of *rational robustness* given by Karpas *et al.*, a social law is considered to be robust if it prevents *any* possibility for a conflict between the agents, whatever plans the agents choose to execute. Formally characterizing it as follows:

**Definition 1.** *A MAP $\Pi$ is robust to rational iff an action sequence that arbitrarily interleaves any individual plans of all agents achieves the individual goals of these agents.*

In what follows, we address the social laws under which a MAP $\Pi$ is robust to rational, or simply *robust*. To this end, we refer to the definition of social laws by Nir *et al.* (2020)

---

[2]Obviously, Karpas *et al.* did not coin the term per se. Various social laws were introduced to multiple other domains much prior to the mentioned work.

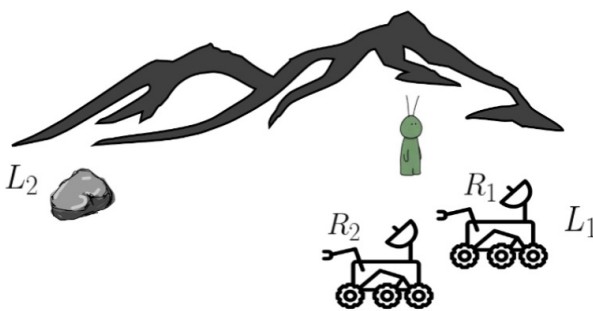

Figure 1: The ROVERS Toy Problem.

**Definition 2.** *For a MAP* $\Pi$ *with total action set A, we call a set of restricted actions* $L \subseteq A$ *a social law. We say that a social law* $L$ *is robust if the restricted MAP* $\Pi \setminus L$ *is robust.*

In other words, a *social law* $L$ is *robust* if it allows *any* scheduling of the agents' plans: preventing any possible conflicts and ensuring that each agent will reach its goal at the end of the execution, i.e., that any scheduling of plans for the modified agent projections $\Pi_i \setminus L$ yields no conflict.

The procedure proposed by Karpas *et al.* to verify that a MAP $\Pi \setminus L$ is robust can be seen as two independent sub-procedures, that both solve classical planing problems:

**IS_SOLVABLE_FORALL**$(\Pi \setminus L)$ checks whether for each agent $i$ the projection $\Pi_i \setminus L$ is solvable, i.e. agent $i$ can reach its goal. We say that a social law $L$ is *feasible* if the procedure returns *true*.

**FIND_CONFLICT**$(\Pi \setminus L)$ ensures that there are no conflicts in any possible schedule. This is achieved by the construction of a classical planning task whose goal is to reach a conflict, where *unsolvability* means that there are no scheduling interferences.

The social law $L$ is considered to be robust if the first sub-procedure returns *true* (the social law is feasible), and the second returns *unsolvable* (there are no conflicts). Here it is important to note that while the unsolvable problems were mostly ignored in classical planning in the previous decades, considerable progress was achieved in the past years (Muise and Lipovetzky 2015).

**Example:**  We illustrate these concepts with a toy problem from the ROVERS domain (see Figure 1) with two locations $L_1, L_2$, two rovers $R_1, R_2$ and, one rock sample to collect. The actions that can be taken by the rovers are *move* – that moves rover between locations and *collect* – collects a rock sample while destroying the rock (the action is irreversible). Suppose that the rock sample is to be taken by $R_1$ rover from $L_2$ and both rovers are initially positioned at $L_1$. For simplicity, the goal of $R_2$ is considered to be an empty set. Two possible robust social laws for this task are: forbid the rover $R_2$ to take the sample $sample(R_2, L_2)$, or forbid the rover $R_2$ to move to location $L_2$, which results in the social law $L = \{move(R_2, L_1, L_2)\}$. Note that if $R_2$ cannot move to $L_2$, it also cannot collect the rock sample located at $L_2$.

## The Search for Social Laws

In their work, Nir *et al.* introduced a forward search approach that returns a robust social law for a given MA-STRIPS problem $\Pi$, if there is one. The problem consists of the following components:

- The original MA-STRIPS problem is considered to be the *initial state* $s_0$. This is the MAP $\Pi$ equipped with the empty social law $L_0 = \emptyset$.

- *The transition function* $\tau(s, a)$, where $s$ is the state associated with the social law $L$ ($s = \Pi \setminus L$) and $a$ is an action of the original MAP $\Pi$. The application of $a$ in the state $s$ results in the state/MAP $s' = \Pi \setminus (L \cup \{a\})$, with the associated social law $L' = L \cup \{a\}$.

- The *goal test* function, as mentioned in the previous subsection, is the test for two independent sub-procedures:

$$\text{IS\_SOLVABLE\_FORALL}(s) \wedge \neg\text{FIND\_CONFLICT}(s)$$

The size of this search space is $2^{|A|}$ where $A$ is set of all actions in $\Pi$, and given a fixed planning task, each state can be seen as a set of the actions forbidden by its social law. Unfortunately, uninformed search algorithms, such as BFS or DFS, do not scale well. Thus, Nir *et al.* proposed various techniques, such as pruning, heuristics and preferred operators identification, to speed up the search. In this paper, we will rely on learning using graph convolutional neural networks, described next, to better guide the search.

## Graph Convolutional Neural Networks

To predict a social law based on previously seen social laws for other MAPs, we need to introduce some compact representation that encapsulates the complex relations between the components of a given MAP. We use a graph representation where facts, actions, and agents correspond to graph vertices, while the relations between them are encapsulated in graph edges. Next, we need a scalable approach for supervised learning on graph-structured data. The method we chose is based on an efficient variant of convolutional neural networks which operates directly on graphs, i.e., Graph Convolutional Neural networks (GCN) (Kipf and Welling 2017). This model scales linearly in the number of graph edges and learns hidden layer representations that encode both local graph structure and features of nodes.

The GCN models we use classify nodes by propagating information through a multi-layer architecture with convolutional layers that capture structural patterns. Specifically, given a graph $\mathcal{G} = (\mathcal{V}, \mathcal{E})$, the network's input is:

- A feature vector $x_v \in \mathbb{R}^d$ for every node $v \in \mathcal{V}$; summarized in a $|\mathcal{V}| \times d$ feature matrix $X$, where $d$ is the feature vector dimension, and

- An adjacency matrix $E \in \{0, 1\}^{|\mathcal{V}| \times |\mathcal{V}|}$ (where the matrix entry $e_{uv} = 1$ if $e_{uv} \in \mathcal{E}$ or $u = v$, and 0 otherwise). Note that $E$ represents the edges of the graph $\mathcal{G}$.

Every layer in this network is written as a non-linear activation function $\sigma$, where

$$H^{(k)} = \sigma(H^{(k-1)}, E), \; \forall k \in \{1, \ldots, K\}$$

with $H^{(0)} = X$ and $H^{(K)}$ the output feature matrix, $K$ being the number of layers. The various GCN models presented in the literature mostly differ in how $\sigma(\cdot, \cdot)$ is constructed and parameterized (Duvenaud et al. 2015; Li et al. 2016; Jain et al. 2016).

In the following sections, we introduce a method that, given some training MAP examples and their corresponding social law, yields a robust social law for unseen examples from the same domain. In other words, given training data: $\{(\Pi_j, L_j)\}_{j=1}^m$, where: $\Pi_j = \langle F, \{A_i\}_{i=1}^n, I, \{G_i\}_{i=1}^n \rangle$ is a MA-STRIPS planning problem with $|A| = k$ being the number of total grounded actions and, $L_j \in 2^k$ is a label vector which denotes which actions are to be restricted by a possible robust social law for this planning problem, it is required to build a model that can formulate a robust social law, i.e. a subset of actions $L \subseteq A$.

## Social Laws: Graph Representation

In the next subsection, we show what is the graph representation of a given MA-STRIPS planning problem and the corresponding social law.

### Problem Description Graphs

The concept of the *problem description graph* (PDG) is not new. In classical planning, PDGs were introduced by Pochter *et al.* (2011) in the setting of symmetry detection for problem formulated in SAS+ (Bäckström and Nebel 1995). Later on Shleyfman *et al.* (2015) rewrote this formulation for STRIPS, and Sievers *et al.* (2019) used this approach for learning a portfolio planner selection. We, however, adopt the PDG for the MA-STRIPS formalism.

**Definition 3.** *Let* $\Pi = \langle F, \{A_i\}_{i=1}^n, I, \{G_i\}_{i=1}^n \rangle$ *be a* MA-STRIPS *planning task. The **problem description graph** (PDG) of* $\Pi$ *is the graph* $\langle V, E \rangle$ *with*

- *Graph vertices* $V = V_F \cup V_A \cup V_{Ag}$, *such that,*

$$V_F = V_F^+ \cup V_F^- = \{v_f^+, v_f^- \mid f \in F\},$$
$$V_A = V_A^{main} \cup V_A^{pre} \cup V_A^{eff} = \{v_a^{main}, v_a^{pre}, v_a^{eff} \mid a \in A\},$$
$$V_{Ag} = \{v^i \mid i \in \{1, \ldots, n\}\}.$$

- *$E$ is the graph edges set:*

$$E = E_F \cup E_A \cup E_{Ag}, \text{ where}$$
$$E_F = \{\{v_f^+, v_f^-\} \mid f \in F\},$$
$$E_A = \bigcup_{a \in A} \left(E_a^{main} \cup E_a^{pre} \cup E_a^{eff}\right), \text{ where}$$
$$E_a^{main} = \{\{v_a^{main}, v_a^{pre}\}, \{v_a^{main}, v_a^{eff}\} \mid a \in A\}$$
$$E_a^{pre} = \{\{v_f^+, v_a^{pre}\} \mid f \in \text{pre}(a)\} \cup$$
$$\{\{v_f^-, v_a^{pre}\} \mid f \in \text{npre}(a)\}$$
$$E_a^{eff} = \{\{v_f^+, v_a^{eff}\} \mid f \in \text{add}(a)\} \cup$$
$$\{\{v_f^-, v_a^{eff}\} \mid f \in \text{del}(a)\}, \text{ and}$$
$$E_{Ag} = \{\{v^i, v_f^+\} \mid f \in G_i\} \cup \{\{v^i, v_a^{main}\} \mid a \in A_i\}$$

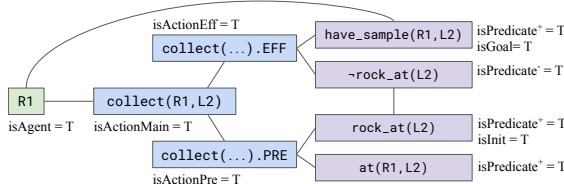

Figure 2: Part of the PDG of Example ROVERS Problem Corresponding

- *Each $v \in V$ has 8 binary attributes as follows:*
  $isPredicate^+(v) = T$ *for* $v \in V_F^+$ *else* $F$
  $isPredicate^-(v) = T$ *for* $v \in V_F^-$ *else* $F$
  $isAgent(v) = T$ *for* $v \in V_{Ag}$ *else* $F$
  $isActionMain(v) = T$ *for* $v \in V_A^{main}$ *else* $F$
  $isActionPre(v) = T$ *for* $v \in V_A^{pre}$ *else* $F$
  $isActionEff(v) = T$ *for* $v \in V_A^{eff}$ *else* $F$
  $isInit(v) = T$ *for* $v \in (V_f^+ \cap I) \cup (V_f^- \setminus I)$ *else* $F$
  $isGoal(v) = T$ *for* $v \in V_f^+ \cap G$ *else* $F$
  *where* $G = \bigcup_{i=1}^n G_i$

Informally, the PDG has three types of vertex clusters: $V_F$ – each fact can be either true or false, and thus represented by two vertices; $V_A$ – actions have three vertices each, the vertices of the action $a$ correspond to the elements of the tuple $\langle \text{pre}(a), \text{npre}(a), \text{add}(a), \text{del}(a) \rangle$; $V_{Ag}$ – each vertex here represents an agent. For example, if $f \in F$ is a predicate such that $f \in I \cap G_j$ for some agent $j$, the following attributes hold true for $v_f$: isPredicate$^+(v_f)$, isInit$(v_f)$, and isGoal$(v_f)$.

The edges indicate the connections between the vertices. Firstly, we connect the vertices of the clusters of actions and facts. Secondly, the precondition and effect vertices of each action are connected to their corresponding facts. Finally, we connect the agents' vertices to their actions and goal facts.

The vertex attributes are introduced to mark vertex type (agent, fact, etc.) and to specify such conditions as being in the initial state or being in one of the agents' goals. Overall, there are 4 attributes for facts, 3 attributes for actions, and 1 attribute to designate agent vertices. It is important to note that all agent vertices have the attributes, therefore the agents are interchangeable, i.e., there is no particular order on the agent that will make one agent more "important" than the other.

Figure 2 shows part of the PDG for our ROVERS toy problem, corresponding to the action collect($R1, L1$). The node $R_1$ represents the first rover agent. The node collect($R_1, L_2$) represents the rover's collect action and, is connected to its corresponding precondition and effect nodes that, in turn, are connected to 4 additional fact nodes.

### Data Representation

In our case, we start with $m$ MAP examples $\{\Pi_1, \ldots, \Pi_m\}$ from the same domain, and a domain-expert. For each of these tasks, the expert produces a social law $L_i$ (we assume

that these social laws do exist), such that the augmented MAP $\Pi_i \setminus L_i$ is robust.

Each task $\Pi_i$ is represented by a graph $PDG(\Pi_i) = \langle V_i, E_i \rangle$, where $E_i \in \{0,1\}^{|V_i| \times |V_i|}$ is the corresponding adjacency matrix, $V_i \in \{0,1\}^{|V| \times 8}$ is the nodes feature matrix, i.e. each row represents a node and its attributes.

The social law is represented as a vector of vertex labels, i.e. $y_i = \{-1, 0, 1\}^{|V_i|}$, where -1 is the label of the non-action vertices, 0 is the label for action vertices that are not included in the social law, and 1 is the label for the action vertices that are included in the robust social law for $\Pi_i$. Using this notation the training data above can be represented as a series of tuples

$$\{(PDG_1, y_1), \ldots, (PDG_m, y_m)\}$$

This data is used to train the GCN-based models. The exact architecture of the GCN is described in Table 1 and, discussed in the Empirical Evaluation section.

## Learning-Based Social Law Synthesis Method

In this section, we introduce a method that utilizes a given dataset, as described above, to train a predictive model. This model, in turn, is exploited to synthesize a robust social law for an unseen MAP.

The output of such a predictor may contain some errors. One of our contributions is a procedure that can handle such inaccuracies. The model may fail on both counts: it can restrict a necessary and completely harmless action, or, on the contrary, allow a conflict-inducing action; in both cases, the predicted social law is not robust.

Thus, it is necessary to establish a method that uses the knowledge from the predictor while guaranteeing the robustness of the synthesized social law. The learning-based social law synthesis method proposed here has three primary steps, which we further describe in the following subsections:

**Train and Predict** given a dataset with $m$ MAPs and their corresponding social laws from a certain MA-STRIPS domain, we train a network to predict a social law for an unseen MAP from the same domain.

**Preprocessing** given a predicted social law for an unseen MAP, the first step we take to find a robust social law is performing some relatively low-cost preprocessing to eliminate some easy-to-find mistakes resulting in a *feasible* social law, i.e., one where every agent can still achieve its goal individually.

**Search for Robust Social Law** after the preprocessing is done, we perform a search through the space of possible social laws, guided by the prediction of our network.

### Train and Predict with GCN based models

Let us first briefly describe how we train and evaluate GCN-based models to predict social laws.

**Network Structure:** All models have 4 or 6 convolutional layers, as described in Table 1, followed by a batch normalization layer, a ReLU activation layer and, a 50% dropout. The network output is passed through a sigmoid function.

| Model | Model Structure | # Parameters |
|-------|-----------------|-------------|
| Net-1 | 4× C128 - sigmoid | 52,098 |
| Net-2 | 4× C256 - sigmoid | 202,498 |
| Net-3 | 3× C128 - 3× C256 - sigmoid | 201,730 |

Table 1: Models' Structures. We use C128 to mark a graph convolutional layer with 128 parameters, a ReLU and a batch normalization layer.

**Training:** First, we randomly split the given dataset to train, validation, and test subsets according to the following ratios: 50%, 25%, and 25% respectively. Then, we train 3 networks (see Table 1) with the Adam optimizer (Kingma and Ba 2015) using negative log-likelihood as the criterion. The training parameters are as follows: two possible learning rates: (1e-4, 5e-5), three possible batch sizes: (2,8,16) and, the default Adam optimizer settings. We also use an early stop mechanism with 50 epochs of patience, testing for improvement in validation set loss.

We are interested in social laws that are permissive, i.e. restrict only a small fraction of the agents' actions, which is why the labels in our dataset are imbalanced – with more allowed actions (false labels) than restricted actions (true labels). Furthermore, as described in the next section, detecting false-positives requires less effort. Thus, the procedure emphasizes recall over precision and gives a higher weight (x3) to false-negative errors. Finally, the procedure picks the best network according to its balanced accuracy score, i.e. the arithmetic mean of the true positive and the true negative rates.

**Prediction:** the predicted social law $L^M$ is the set of all graph nodes (actions) that got a positive prediction by the network, i.e., a score higher than 0.5. Let us define $M : A \rightarrow [0,1]$, a function that maps an action node to its network's score. Thus, the predicted social law $L^M$ is the set of all actions that got a positive score $L^M = \{a \in A \mid M(a) > 0.5\}$. As previously mentioned, enacting $L^M$ on $\Pi$ gives a similar multi-agent problem only without the actions included in $L^M$.

Note that $L^M$ is not guaranteed to be robust thus, the purpose of the following steps is to detect and correct possible errors in $L^M$, to produce a robust social law.

### Preprocessing of the Predicted Social Law

Here we describe a relatively low-cost procedure to process the predicted social law and fix possible errors. The procedure goes through $L^M$ and detects what restrictions prevent agents from achieving their goal individually, making $L^M$ an *infeasible* social law. Then, to establish a feasible social law, the procedure removes these parts from the social law, that is, it allows these actions. Let us note that, while this preprocessing procedure ensures that the produced social law is feasible, there still are no guarantees about its robustness.

The procedure consists of two main steps: (a) checking every single-action restriction in $L^M$ individually and removing the restrictions that make it infeasible, and (b) per-

forming a binary search through the remaining subsets of the social law, aiming to find one that is feasible, trying to keep the majority of $L^M$. In the following, we further describe the above-mentioned steps.

We implement the first step by constructing $MSL = \{L_1, L_2, \ldots, L_n\}$ a set of $n$ atomic (one-sized) social laws s.t. each $L \in MSL$ consists of only one action restriction from $L^M$. Then, for every $L \in MSL$, the procedure confirms that the affected agent can achieve its goal, i.e. $L$ is feasible, by solving the single-agent projection of the planning problem. We denote these safe restrictions as *applicable* restrictions. We finish the first step by constructing $L_r^M \subseteq L^M$, the refined social law that consists of all applicable restrictions from the predicted social law. In other words, we discard from $L^M$ actions which, if we include them in the social law, will prevent one of the agents from achieving its goal individually.

Despite the first step, $L_r^M$ may be an infeasible social law, as restricting two actions could prevent some agent from achieving its goal, even if restricting each of these separately does not. Of course, checking all possible subsets of $L_r^M$ is not tractable so, to acquire some feasible social law, we introduce the second step of the procedure that executes a binary search to remove parts of $L_r^M$ such that the resulting social law is feasible. The intuition behind this step lies in the assumption that the actions with a lower $M(a)$ score (that is, where the network is less certain) should be restricted with lower priority. Thus, in case of an infeasible social law, the procedure would prefer to discard these actions first.

The binary search procedure is implemented by, first, constructing a list $l = [a_1, a_2, \ldots]$ with the actions in $L_r^M$, sorted by $M(a)$, i.e., the higher score actions (the actions that should be restricted according to the network's prediction) are at the suffix of the list. Then, the procedure performs a binary search in $l$ to detect the largest contiguous suffix s.t. the corresponding social law is feasible, i.e., all agents can achieve their goals — this is verified by calling IS_SOLVABLE_FORALL for every suffix that is checked. We denote the resulting social law as $L_f^M$ – the feasible predicted social law. Let us note that this step could result in an empty social law.

It is easy to see that $L_f^M$ is a feasible social law for $\Pi$ as the second phase confirms it before producing it. Also, our empirical evaluations shows that this procedure scales well and can produce large ($\sim 150$ actions) and feasible social laws in less than a minute.

### Searching for a Robust Social Law

In the previous section, we described a procedure that fixes one type of mistake in the predicted social law $L^M$, producing a feasible social law $L_f^M \subseteq L^M$. However, we may need to add necessary and more complex modifications to $L_f^M$ to obtain a robust social law. Here we describe a method, based on the search technique from Nir *et al.*, to tackle this challenge.

We adapt the social law search technique from Nir *et al.* to correct the remaining errors, if there are any, in $L_f^M$. Most of the search problem settings, as briefly described in the

| Domain | Empty Open List | | | Modified Open List | | | Base |
|--------|-----------------|-----------------|-----------------|--------------------|-----------------|-----------------|------|
|        | $h_{\text{sum}}$ | $h_{\text{avg}}$ | $h_{\text{cnt}}$ | $h_{\text{sum}}$ | $h_{\text{avg}}$ | $h_{\text{cnt}}$ |      |
| ROVER  | 6  | 4  | 5  | 9  | 10 | 10 | 4  |
| ZENO   | 10 | 10 | 10 | 10 | 10 | 10 | 4  |
| TAXI   | 4  | 4  | 4  | 4  | 4  | 4  | 4  |
| DLOG   | 7  | 7  | 4  | 9  | 7  | 7  | 4  |
| TOTAL  | 27 | 25 | 23 | **32** | 31 | 31 | 16 |

Table 2: Number of Successful Searches on IPC Benchmarks (size of train set = 64, noise = 0%, ZENO = ZENO-TRAVEL, DLOG = DRIVERLOG, Baseline = GBFS-$h_{stat}$-NP from (2020))

preliminaries, remain the same as well as the search safe pruning techniques.

Additionally, we suggest 2 ways to use the predictor function. First, we suggest inserting $L_f^M$ to the open list at the beginning of the search, as we assume it will be closer to a robust social law than the standard initial social law, $\emptyset$. We remark that the empty social law is also put on the open list, so our method remains complete.

Second, we introduce 3 new heuristic functions which rely on the social law predictive function $M : A \to \mathbb{R}$ to guide the search. Let us note that our greedy search is set to explore nodes (social laws) with higher heuristics value.

**Sum of Scores Heuristic** $h_{\text{sum}}(L) = \sum_{a \in L} M(a)$

This heuristic sums $M(a)$ for all actions restricted by the social law ($a \in L$), making the search explore social laws that restrict high-score actions, i.e. actions that, according to the network, are more likely to be included in a robust social law. Note that this heuristic will always prefer to add more actions to the social law, without regard to its size.

**Average Score Heuristic** $h_{\text{avg}}(L) = \frac{1}{|L|} \sum_{a \in L} M(a)$

This heuristic computes the arithmetic mean of $M(a)$ restricted by the social law, and thus guides the search towards smaller, less-restrictive social laws. However, its preference for smaller social laws that include only high-score actions may limit its use when handling larger problems.

**Count Heuristic** $h_{\text{cnt}}(L) = |\{a \in L \mid M(a) > 0\}|$

This heuristic returns the number of restricted actions with a positive $M(a)$ score. One may use this heuristic when $M(a)$ is noisy and the true robust social law may need to restrict actions that got a negative score.

## Empirical Evaluation

We begin with a description of our data generation procedure, and then evaluate our approach on problems from Nir *et al.* (2020) that are based on domains from the first Competition of Distributed and Multi Agent Planners (Komenda, Stolba, and Kovacs 2016).

### Data Generation

To evaluate our approach, we need a way to generate robust social laws for multiple problems in each domain we use. Therefore, we implemented automatic social law generators for 4 domains: DRIVERLOG, ZENOTRAVEL, TAXI and,

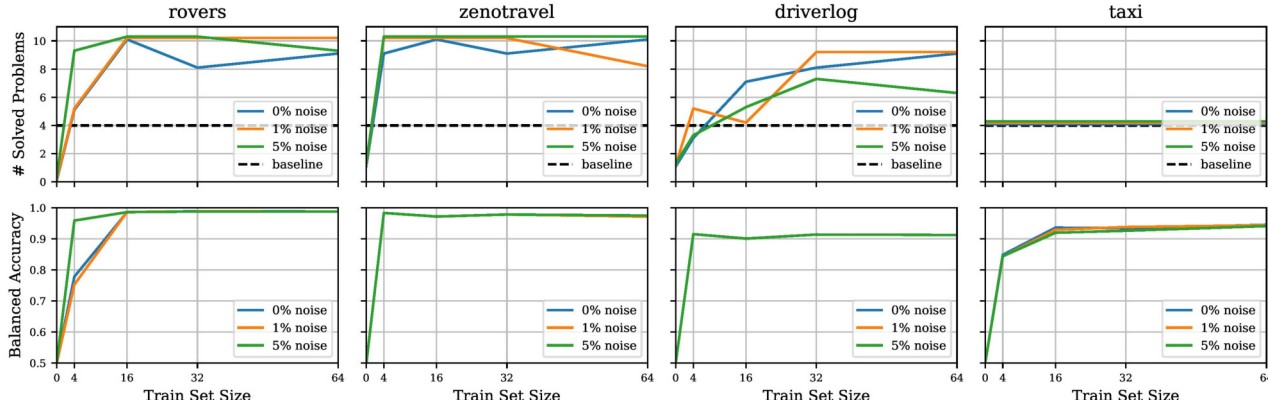

Figure 3: Number of Successful Searches on IPC Benchmarks with Varying Training Set Sizes (using $h_{\text{sum}}$-MOL search configuration, baseline values are based on GBFS-$h_{stat}$-NP search configuration (2020))

ROVERS, for which we also have random problem generators. These social law generators are meant to simulate a domain-expert, but are not perfect – we implemented them based on a small number of problems, and in some cases they do not generalize correctly to other randomly generated problems. Thus, they simulate an imperfect expert. As we will see, our technique is still able to use the labels from such an imperfect expert to generate robust social laws.

This setup simulates a single expert who applies the same class of social law to problems from a subset of a multi agent planning domain. It is true that if multiple experts were to provide social laws for different problems in the same domain, they could formulate different classes of social laws – this is left for future work. Note that although we manually wrote these social law generators for each domain, for the purpose of the empirical evaluation, the social law learning technique we present and evaluate is domain-independent – the same technique is used in all domains.

In addition to the imperfect labeling mentioned above, we wanted to simulate and control additional errors. One type of error a domain-expert might make is needlessly restricting some actions, leading to an overly restrictive social law. To simulate this in our empirical evaluation we added artificial noise to the social law generator output. We implement this by randomly adding some percentage of the agents' actions to the generated social law. Note that, mistakes that make a social law not-robust can be detected using social law robustness verification (Karpas, Shleyfman, and Tennenholtz 2017) thus, the noise we induce is unidirectional, and affects only allowed actions. Additionally, the number of the allowed actions is usually much smaller than the size of the restricted actions so adding only 5% noise can actually double the size of the social law.

### Robust Social Law Synthesis

We have described three different network-based heuristics $h_{\text{sum}}$, $h_{\text{cnt}}$ and, $h_{\text{avg}}$, as well as the technique for seeding the open list with the predicted social law. Thus, we have 6 different greedy search configurations to compare: using one out of the three heuristics, either with open list modification

(denoted MOL) or without (denoted NM). These are tested on 80 planning problems from 4 different domains.

Similarly to previous work on automated social law synthesis, the search goal test calls the *Fast Downward Stone Soup* (FD) (Röger, Pommerening, and Seipp 2014) and *SymPA* (Torralba 2016) planners for the goal test – checking if the current social law is robust. We too deploy a sequential order of timed attempts (2-minute attempts (FD and SymPA) followed by 30-minute attempts (again, FD, then SymPA)).

We consider a search successful if it either found or disproves the existence of a robust social law. Table 2 reports the number of successful searches in each domain with 0% noise and 64 training samples. The baseline – the best social law synthesis configuration from the previous work (2020) – was able to solve 16 of these problems. All of the proposed configurations solved more problems. These results show that inserting the predicted social law to the search open list (denoted MOL) leads to at least 5 more successful searches. Our best configuration, using $h_{\text{sum}}$ with MOL solves twice as many problems as the baseline.

The results in the TAXI domain show that there is no robust social law present for all of the solved problems. In fact, we suspect that this result holds for the rest of the domain. As expected, this shows that our approach does not have an advantage in proving a robust social law does not exist.

Next, we evaluate the effects of the size and quality of the training set. Figure 3 shows the number of problems solved ($y$ axis of the plots in the top row) and the balanced accuracy score ($y$ axis of the plots on the bottom row) in each domain with various noise levels, for varying sizes of the training set ($x$ axis). We calculate the balanced accuracy score on an unseen test set of generated problems and their corresponding social law. Putting aside TAXI, the results show that 4 training samples are enough to beat the baseline that is marked on the graph as a horizontal dashed line. Note that the trends in the graphs point that there is a correlation between the number of successful searches and the balanced accuracy score.

Interestingly, a limited amount of noise does not seem to hurt and, in fact, our results show that noise may sometimes help the network in achieving higher balanced accuracy and

| | Search Time | | | | | | Social Law Length | | | | | |
|---|---|---|---|---|---|---|---|---|---|---|---|---|
| Instances | Empty Open List (NM) | | | Modified Open List (MOL) | | | Empty Open List (NM) | | | Modified Open List (MOL) | | |
| | $h_{\text{sum}}$ | $h_{\text{avg}}$ | $h_{\text{cnt}}$ | $h_{\text{sum}}$ | $h_{\text{avg}}$ | $h_{\text{cnt}}$ | $h_{\text{sum}}$ | $h_{\text{avg}}$ | $h_{\text{cnt}}$ | $h_{\text{sum}}$ | $h_{\text{avg}}$ | $h_{\text{cnt}}$ |
| ROVERS-0 | 183.75 | 193.07 | 188.42 | 123.01 | 1931.3 | 122.92 | 15 | 15 | 15 | 16 | 16 | 16 |
| ROVERS-1 | 181.84 | 180.99 | 179.98 | 123.41 | 126.83 | 123.3 | 12 | 12 | 12 | 14 | 14 | 14 |
| ROVERS-3 | 248.23 | 274.47 | 247.97 | 127.2 | 1930.56 | 127.44 | 18 | 18 | 18 | 20 | 20 | 20 |
| ROVERS-5 | 174.99 | 215.17 | 1984.85 | 126.81 | 149.26 | 124.41 | 7 | 8 | 8 | 11 | 11 | 11 |
| ZNTL-0 | 140.99 | 129.61 | 129.29 | 30.02 | 20.48 | 16.03 | 5 | 4 | 5 | 22 | 22 | 22 |
| ZNTL-1 | 43.58 | 103.44 | 130.49 | 19.52 | 20.48 | 17.32 | 6 | 5 | 4 | 22 | 22 | 22 |
| ZNTL-2 | 1860.55 | 134.9 | 145.19 | 121.92 | 131.13 | 121.86 | 4 | 4 | 4 | 28 | 4 | 28 |
| ZNTL-3 | 141.78 | 1949.14 | 156.73 | 121.95 | 136.01 | 121.99 | 7 | 6 | 6 | 36 | 36 | 36 |
| ZNTL-4 | 137.37 | 138.25 | 137.39 | 753.75 | 134.51 | 121.93 | 5 | 5 | 5 | 36 | 5 | 36 |
| ZNTL-5 | 192.89 | 194.52 | 191.74 | 123.39 | 128.22 | 123.41 | 14 | 13 | 13 | 87 | 87 | 87 |
| ZNTL-6 | 216.22 | 217.37 | 235.15 | 123.56 | 211.48 | 123.48 | 14 | 14 | 16 | 98 | 14 | 98 |
| ZNTL-7 | 248.3 | 232.68 | 243.03 | 124.54 | 213.19 | 123.33 | 16 | 16 | 16 | 112 | 16 | 112 |
| ZNTL-8 | 2071.7 | 274.92 | 299.67 | 124.66 | 263.31 | 1928.34 | 16 | 16 | 19 | 137 | 16 | 137 |
| ZNTL-9 | 309.3 | 2125.22 | 319.14 | 124.73 | 309.57 | 1928.5 | 18 | 18 | 18 | 153 | 18 | 153 |
| TAXI-0 | 7.81 | 7.88 | 7.85 | 10.79 | 10.58 | 10.27 | NSL | NSL | NSL | NSL | NSL | NSL |
| TAXI-1 | 13.51 | 13.95 | 13.6 | 13.68 | 15.26 | 16.79 | NSL | NSL | NSL | NSL | NSL | NSL |
| TAXI-2 | 32.36 | 12.42 | 14.35 | 16.68 | 17.74 | 17.58 | NSL | NSL | NSL | NSL | NSL | NSL |
| TAXI-6 | 39.72 | 36.54 | 32.09 | 17.97 | 19.81 | 18.13 | 7 | 7 | 7 | 37 | 37 | 37 |
| DLOG-0 | 10.19 | 10.24 | 7.7 | 1.22 | 2.29 | 1.25 | 4 | 4 | 5 | 35 | 35 | 35 |
| DLOG-3 | 147.4 | 362.76 | 689.15 | 119.32 | 126.74 | 127.13 | 11 | 9 | 9 | 47 | 78 | 78 |
| AVG | 320.1 | 340.4 | 267.7 | 117.4 | 294.9 | 265.8 | 10.5 | 10.2 | 10.6 | 53.6 | 26.5 | 55.4 |

Table 3: Search Time on IPC Benchmarks (Right), and Length of the Resulting Social Laws (Left) (size of training set = 64, noise = 0%, NSL = robust social law does not exist, ZNTL = zenotravel domain)

solving more problems. Note that adding noise to training data is considered to be a standard data augmentation technique in deep learning because it expands the size of the training dataset. However, in our procedure, it is not the case because the noise is encoded in training samples. Of course, we do not add noise to the test set samples.

All in all, 20 problems were solved by all of the configurations and are used to compare search times and social law length as reported in Table 2. As our results show, using MOL improves search times, mostly when using $h_{\text{sum}}$ and $h_{\text{avg}}$. The most likely explanation for that is that the predicted social law is closer to the search goal than an empty social law, and thus the search can be shallower. However, using MOL tends to result in more restrictive social laws. This may be handled by employing $h_{\text{avg}}$ that, as expected, may be slower but tends to regulate the length of the resulted social law.

## Conclusion

In this paper, we explored the use of graph neural networks to better guide the process of social law synthesis. Our empirical evaluation shows that the proposed techniques are scalable; we solve twice as many problems as the previous state-of-the-art. Furthermore, the results show that these techniques can learn from an imperfect expert, even in the presence of noise.

Given the PDG representation we used is domain-independent, we see two interesting directions for future research: adding domain-specific features (predicate name for fact nodes, operator name for action nodes, etc.) and, transfer learning between domains that exploits possible domain structure similarities.

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
