# OpenReview forum: "Learning-based Synthesis of Social Laws in STRIPS"
_icaps-conference.org/ICAPS/2021/Workshop/HSDIP — HSDIP 2021_

### Official Review · AnonReviewer2 · 2021-05-26

**Confidence:** 4
**Overall Score:** Strong Accept

**Review:**

**Tittle: Learning-based Synthesis of Social Laws in STRIPS**

### Summary
In Multi-Agent Problems, a social law prohibits some agents to execute some actions. A social law is *robust* if it guarantees that all agents will reach their goal. Finding good social laws seems to be tricky.
The authors train a Graph Convolutional Network (GCN) to predict social laws (or more precise to predict a score for every action). As always, neural networks (NN) provide no guarantees, thus, the GCN is used to seed the search for a robust social law and as a heuristic. The search algorithm gives us the desired guarantees.


### Feedback
I really like your work. With some background on GCNs, it is fairly easy to read. Except for some minor, mostly stylistic issues (see below), I do not have much to criticize.
Table 3 contains a lot of information which make it very hard to read. Thus, the reader quickly moves to the last row for the averages. This is problematic, because important information could be skipped. For example, all except for one configurations have 1-2 outlier with respect to time. The outliers cannot be seen in the average row and you do not speak about them. I hope you can somehow restructure it. E.g. for the right block you could add the baseline (empty open list, no heuristic) as first column and the other columns contain only +x or -x to denote the change.


### Questions
- *Table 3:* Where do the search time outliers come from?
- *Empirical Evaluation, Data Generation:* "The number of allowed actions is usually much smaller than the size of the restricted actions, so adding only 5% of noise can actually double the size of the social law.". Previously you state that you can produce large social laws (~150 actions). If this is much larger than the allowed actions, do your state space contains just 200 actions or do I misunderstand something?
- *Table 1:* The table shows for the first row that the model has 4 layers with 128 parameters, but at the same time it shows that the model has 52,098 parameters in total. What do you mean with 128 parameters? How do these numbers relate?
- *Train and Predict with GCN base models, Train:* You weight false-negative errors thrice. Did you just chose this value or is there a justification, except for an intuition, behind it (e.g. preliminary experiments)?
### Minor Issues
- *Preliminaries:* "and a given arbitrary scheduler executes the actions one by one with no given priorities, " This confused me a lot. I was also thinking about no priorities between actions of the same plan. Definition 1 cleared it up. Maybe reformulate this and use the wording with interleaving that was used below.
- *Social Laws: Graph Representation, node attributes:*
	- "if" -> "for"; my first association was why should we iterate now over the set of nodes? Afterwards, I still think it the text contains a lot of redundancy which makes it harder than necessary to read.
	- Alternative: Each $v \in V$ has the following 8 binary attributes. We define $x\stackrel{?}{\in}X$ as true if $x\in X$ and false otherwise.
		- $isPredicate^+(v) = v\stackrel{?}{\in}V^+_F$
		- ...
	- If each line becomes shorter, then aligning at the "=" symbol could further improve the readability.
- *Preprocessing of the Predicted Social Law:*
- *Table 2:* I would move the baseline column to the front, because my first intuition was to see how good is the baseline in this domain and then compare it to the new approaches.
- *Table 3:*
	- Right and Left are switched in the caption.
	- I would move the "(Right)", "(Left)" annotations prior to the phrase they belong to
	- When you speak about table 3, you have written table 2 (last paragraph prior to conclusion).

---

### Official Review · AnonReviewer1 · 2021-05-28

**Confidence:** 4
**Overall Score:** Accept

**Review:**

This paper introduces a new approach for synthesizing Social Laws for unseen situations by
taking advantage of previous experience in the same domain. The approach constructs a
heuristic function by using Graph Neural Networks and shows how such a heuristic can guide
a search for constructing a provably robust social law. The experiments show the benefits
of the new heuristic function where compared against previous work.

The paper is well written, with a well-motivated problem and an interesting solution. The
empirical evaluation is quite convincing, analyzing a number of different heuristics
accross several domains, showing that learning is possible in a number of domains.

One question I had while reading the multiple heuristics, that take the sum, average or
count is whether one could combine several of these metrics (e.g. using a multi-queue
search) or just apply learning to select how to best aggregate the evaluations in each domain.

Just as a suggestion about something that could be interesting to look at in future
work. When you mentioned "Of course, checking all possible subsets of L^M_r is not
tractable", this reminded me of the problem of oversubscription planning, and reminded me
of the work by Rebecca Eifler et al.  (AAAI'20/IJCAI'20) and David Speck et al. (AAAI'21),
which have significantly advanced the state of the art. I wonder if their techniques could
be used to improve the preprocessing method to find a minimal (or close to minimal)
feasible social law.


Minor comments:

page 2, is be denoted -> is denoted
page 6, consists all -> consists of all
page 7, of a robust social *law*.
page 7, All of configurations -> the configurations

---

### Author Response · Authors · 2021-05-31
**Authors response**

First, we would like to thank the reviewers for their insightful and helpful comments.
We will consider them all when preparing the final version of this paper.

We now answer the reviewers’ detailed questions, below:

AnonReviewer1:
(1) We agree, combining the different heuristics is an interesting concept and should improve our results, it can be easily addressed by using a multi-queue approach. However, learning-based approaches may be more challenging to implement in this case, we intend to give it more consideration.

(2) Formulating‌ ‌the‌ ‌problem‌ ‌of‌ ‌finding‌ ‌the maximally restrictive feasible social law as an oversubscription planning problem with the hard goal of the law being feasible and the soft goal of restricting as many actions from L^M as possible is an interesting way to approach this problem. We will examine this further, thank you.

AnonReviewer2:
(1) ‌ The ‌causes of outliers are probably related to the disadvantages that heuristics suffer from (as described in the “Searching for a Robust Social Law” section). A more in-depth analysis of a specific outlier case may be helpful.
(2) We agree, there’s a formulation error in that sentence. It should be: “The number of allowed actions is usually much greater than…”
(3) “128” indicates a layer's dimension, not the number of the network’s weight parameters. Taking a simple fully-connected network with a 64-dimension input and a 128-dimension output as an example, there are 64*128 parameters (8,192 ‌weight‌ ‌parameters‌ ‌+‌ ‌1‌ bias ‌ ‌ parameter).
(4) You're right, we did some preliminary experiments that showed putting a higher weight on false-negative errors improves the overall process.
Your further suggestions will be taken into consideration, thank you for that.

Given all of the above, we appreciate the hard work done by the reviewers and believe that their suggestions will improve the paper presented.

---

### Decision · Program_Chairs · 2021-06-10

**Decision:**

Accept

**Comment:**

Congratulations, all reviewers agreed that this is a clear accept.